

# The association between Geographic Information System-based neighborhood built environmental factors and accelerometer-derived light-intensity physical activity across the lifespan: a cross-sectional study

Sofie Compernolle[1,2], Lieze Mertens[1,3], Jelle Van Cauwenberg[2,3], Iris Maes[1] and Delfien Van Dyck[1]

[1] Department of Movement and Sports Sciences, Ghent University, Ghent, Belgium
[2] Research Foundation Flanders, Brussels, Belgium
[3] Department of Public Health and Primary Care, Ghent University, Ghent, Belgium

Corresponding author
Sofie Compernolle,
sofie.compernolle@ugent.be

## ABSTRACT

**Background**. Evidence on associations between environmental factors and accelerometer-derived light-intensity physical activity (LPA) is scarce. The aim of this study was to examine associations between Geographic Information System (GIS)-based neighborhood built environmental factors and accelerometer-derived LPA, and to investigate the moderating effect of age group (adolescents, adults, older adults) on these associations.

**Methods**. Objective data were used from three similar observational studies conducted in Ghent (Belgium) between 2007 and 2015. Accelerometer data were collected from 1,652 participants during seven consecutive days, and GIS-based neighborhood built environmental factors (residential density, intersection density, park density, public transport density, entropy index) were calculated using sausage buffers of 500 m and 1,000 m around the home addresses of all participants. Linear mixed models were performed to estimate the associations.

**Results**. A small but significant negative association was observed between residential density (500 m buffer) and LPA in the total sample (B = −0.002; SE = 0.0001; p = 0.04), demonstrating that every increase of 1,000 dwellings per surface buffer was associated with a two minute decrease in LPA. Intersection density, park density, public transport density and entropy index were not related to LPA, and moderating effects of age group were absent.

**Conclusions**. The small association, in combination with other non-significant associations suggests that the neighborhood built environment, as classically measured in moderate-to-vigorous intensity physical activity research, is of limited importance for LPA. More research is needed to unravel how accelerometer-derived LPA is accumulated, and to gain insight into its determinants.

## BACKGROUND

Physical inactivity is considered to be a major public health issue worldwide due to its contribution to a range of non-communicable diseases and premature mortality (*Katzmarzyk et al., 2021*). Research has estimated that a lack of physical activity causes 6% of the burden of disease from coronary heart disease, 7% of type 2 diabetes, 10% of breast and colon cancer, and 9% of premature mortality, which corresponds to approximately 5.3 million deaths worldwide (*Lee et al., 2012*). To reduce the burden of physical inactivity, updated physical activity guidelines were proposed in 2020 by the World Health Organization (WHO). These updated guidelines emphasize that all types and intensities of physical activity count, bringing greater attention to the previously ignored light-intensity physical activities (LPA) (*Van der Ploeg & Bull, 2020*). According to Ainsworth's Compendium of Physical Activities, LPA's include all activities performed at 1.6–3 metabolic equivalent of tasks (METs) (*Ainsworth et al., 2000*). Examples of frequently performed LPA's are standing, slow walking ($\leq 3$ km/h) and gardening (*Mendes et al., 2018*).

Although often overlooked, LPA can play a key role in health promotion and disease prevention. Recent systematic reviews and meta-analyses have indicated that LPA was favorably associated with some cardiometabolic risk factors including waist circumference, triglyceride levels, insulin, and the presence of metabolic syndrome in different age groups (*Chastin et al., 2019*; *Amagasa et al., 2018*; *Poitras et al., 2016*). LPA was also inversely associated with all-cause mortality risk after adjustment for moderate-to-vigorous intensity physical activity (MVPA) (*Amagasa et al., 2018*; *Ku et al., 2020*; *Fuezeki, Engeroff & Banzer, 2017*). Results of a longitudinal cohort study among 4,840 US adults showed that those adults who performed 4 h/day of LPA had a 21% lower risk of mortality compared with those who did less LPA (*Matthews et al., 2016*). Focusing on LPA might thus be important, particularly for whom the recommended 60 min/day (adolescents) or 150–300 min/week (adults and older adults) of MVPA are challenging to achieve (*Del Pozo Cruz et al., 2021*).

According to the ecological model of active living, physical activity is influenced by individual characteristics as well as characteristics of the environment in which the behavior is performed (*Sallis et al., 2006*). Various studies have addressed which neighborhood built environmental factors promote or hinder more intense levels of physical activity in different age groups (*e.g.*, *Cain et al., 2021*; *De Bourdeaudhuij et al., 2015*; *Lee, Ewing & Sesso, 2009*; *McCormack & Shiell, 2011*). Residential density, intersection density, public transport density, park density and land-use mix (often measured using an entropy index) were frequently identified as correlates of moderate-to-vigorous physical activity or active transport (*McCormack & Shiell, 2011*; *Sallis et al., 2016*; *Van Cauwenberg et al., 2018*). Residential density may impact on physical activity by an increased sense of safety or by increasing access to shops, services and public transport facilities that, in turn, may reduce car dependency and, thus, promote active transport (*Forsyth et al., 2007*; *Cerin et al., 2020*). Intersection density, as an indicator of street connectivity, is expected to influence physical activity by an increase in active transport, possibly because distances to destinations tend to decrease (*Sallis et al., 2016*; *Bungum et al., 2009*). Similarly, dense public transport services
are expected to stimulate active transport, as people usually walk or cycle to the nearest public transport stop (*Sallis et al., 2016*). Parks density may impact physical activity directly as parks are supportive environments for recreational physical activity; and indirectly, as nearby parks can also be a destination for active transportation (*McCormack et al., 2010*). Finally, a higher land-use mix should increase the availability of destinations within walking distance from home, thus decreasing passive transport (*Sallis et al., 2016*).

Importantly, the built environmental factors presented above were mainly examined as correlates of MVPA. Little determinant studies have been conducted focusing on LPA, and the ones that exist have mainly focused on walking (*McCormack & Shiell, 2011*; *Haselwandter et al., 2015*; *Durand et al., 2011*). Walking, which is a common form of LPA if performed slowly, is influenced by one's built environmental neighborhood (*Saelens & Handy, 2008*). Concretely, systematic reviews revealed strong evidence for positive associations between neighborhood walkability, number of destinations, greenery, and aesthetically pleasing environments on the one hand, and walking on the other hand, in different age groups (*McCormack & Shiell, 2011*; *Saelens & Handy, 2008*; *Barnett et al., 2017*; *Smith et al., 2017*; *Wang et al., 2016*). However, important to note, is that the majority of these findings are based on self-reported walking data without taking into account walking speed. The proportion of slow walking in previous studies remains thus unclear, and it might be that only a small part of the self-reported walking can be classified as LPA. In addition, many other LPA's, such as gardening and cooking, are not taken into account when analyzing self-reported walking behavior. Consequently, research into objective accelerometer-derived LPA can provide useful and complementary insights.

A recent systematic review, focusing on objectively Geographic Information System (GIS)-measured built environmental factors and accelerometer-derived physical activity, showed that among the sixty included studies, only eight investigated the influence on LPA (*Yi et al., 2019*). Results of these few studies were also inconsistent (*Whitaker et al., 2019*). For example, in the cross-sectional study of *Chen et al. (2019)* no significant associations were found between neighborhood walkability attributes and LPA in 124 older Taiwanese adults, whereas in the study of *Van Holle et al. (2014a)* and the study of *Gonçalves et al. (2017)* walkability and residential density were inversely associated with LPA in a sample of 438 Belgian older adults, and 305 Brazilian adults, respectively. Next to the potential moderating role of geographical location, the detected inconsistencies might also be attributed to age-related differences in the studied samples.

Neighborhood built environmental factors might be less important for adults' LPA, as this population subgroup generally spend not as much time in their neighborhood as adolescents and older adults due to two reasons (*Glass & Balfour, 2003*). Firstly, many adults are in full-time employment, often outside their own neighborhood, which reduces the available free time in their neighborhood (*Glass & Balfour, 2003*; *Kaczynski et al., 2009*). Secondly, most adults own a car, and have a driver license, which gives them easy access to more distant neighborhoods (*Simons et al., 2017*). In contrast, many adolescents and older adults cannot travel independently by car due to the lack of a driver license, or due to aging-related health issues (*Beard et al., 2009*; *Oyeyemi et al., 2019*; *Vilhjalmsdottir et al., 2016*; *Shigematsu et al., 2009*). Consequently, they are often dependent on public transport

for trips outside their neighborhood. The use of public transport involves several barriers, especially for older adults. A recent study concluded that these barriers can be classified as physical (*e.g.*, struggling to board the bus), emotional (*e.g.*, worrying about getting a seat), and spatiotemporal (*e.g.*, staying home when the weather is bad) barriers (*Ravensbergen et al., 2021*). Next to the importance of the neighborhood, the sausage buffer representing ones neighborhood might also differ across age groups. It is hypothesized that the sausage buffer should be smaller in older adults as their walking distance is more likely to be diminished due to functional limitations (*Cerin et al., 2017*; *Carlson et al., 2012*).

Up till now, existing studies on the association between built environmental factors and physical activity have mainly focused on one specific age group. Although these studies yield interesting insights, they fail to examine the moderating role of age group on the associations between GIS-based neighborhood built environmental factors and accelerometer-derived LPA. However, as the built environment affects all population subgroups living in a certain neighborhood, examining the moderating role of age groups is important to inform policy makers and urban planners on the environmental changes that may yield the greatest benefits for diverse age groups.

To add to the limited knowledge on the determinants of objectively measured LPA, and to inform policy makers and urban planners on health-enhancing environments, the aims of the current study were (1) to investigate the associations between GIS-based neighborhood built environmental factors and accelerometer-derived LPA, and (2) to examine the moderating effect of age group (adolescents, adults, older adults) on these associations. It was hypothesized that stronger associations will be detected between neighborhood built environmental factors and LPA in adolescents and older adults, compared to young and middle-aged adults.

## MATERIALS AND METHODS

### Study design

Data from three observational studies with a similar methodology were combined to examine associations between neighborhood built environmental factors and LPA across the lifespan. All three studies were conducted in Ghent, Flanders. The Belgian Environmental Physical Activity Study (BEPAS) collected data from adults (20–65 years) between May 2007 and September 2008 (*Van Dyck et al., 2010*), and the BEPAS Seniors collected data from older adults (≥65 years) between October 2010 and September 2012 (*Van Holle et al., 2014b*). The Belgian International Physical activity and the Environment Network (IPEN) study in Adolescents collected data from adolescents (11–17 years) between September 2014 and December 2015 (*Cain et al., 2021*). All three studies were approved by the Ethics Committee of the Ghent University Hospital (B670201423000) and all participants provided written informed consent.

### Participant selection and procedure

Stratified cluster sampling based on walkability (*i.e.,* a composite measure of residential density, street connectivity and land use mix diversity; low *vs* high) and median annual household income (as a measure for neighborhood socio-economic status [SES]; low *vs*

high) was used to select neighborhoods (*i.e.,* 1 to 5 adjacent statistical sectors) in Ghent (*i.e.,* city in Flanders, Belgium) for the three observational studies (*Van Dyck et al., 2010*; *Van Holle et al., 2014b*). A total of 24 neighborhoods were selected from four neighborhood types (*i.e.,* high walkable/high SES; high walkable/low SES; low walkable/high SES; low walkable/low SES) to recruit adult participants for BEPAS (*Van Dyck et al., 2010*). Low and high walkable neighborhoods were defined as neighborhoods in the lowest and highest walkability quartiles, respectively. Low and high SES neighborhoods were neighborhoods from the second, third or fourth *versus* seventh, eighth, or ninth decile of neighborhood income. The first and tenth deciles were excluded in order to avoid extremely poor or wealthy neighborhoods skewing our results. Subsequently, 250 adults of each neighborhood were randomly sampled by the Public Service of Ghent. For BEPAS Seniors, 20 out of these 24 neighborhoods were selected to randomly sample 1750 older adults stratified by age and gender (*Van Holle et al., 2014b*). For IPEN Adolescents, 442 adolescents were randomly sampled from the 24 neighborhoods that were initially selected for BEPAS. Next to the recruitment by residential address, adolescents were also recruited from schools located in the 24 neighborhoods (*Cain et al., 2021*). Selected adolescents, adults and older adults received an invitation letter with the announcement of a home or school visit of a trained researcher within the next days. Candidates were considered to be eligible for the study if they lived in a private dwelling, were able to walk a couple of hundred meters without assistance and were able to fill out a Dutch questionnaire. The recruitment process resulted in a sample of 373 adolescents, 1,200 adults, and 508 older adults who were found at home/school, met the inclusion criteria, and willing to participate. All participants filled in a questionnaire on sociodemographic and psychosocial factors, and physical activity. Additionally, one of the parents of each adolescent participant also completed a brief socio-demographic questionnaire. By the end of the first home/school visit, participants received an Actigraph accelerometer, which they were instructed to wear for seven consecutive days. After seven days, a second home/school visit took place to collect the Actigraph accelerometers.

## Measures

### Outcome variable: LPA

LPA was objectively assessed with ActiGraph 7164, GT1M, GT3X and GT3X + accelerometers (ActiGraph, Fort Walton Beach, FL, USA), which are valid and reliable tools to measure PA levels in different age groups (*Aadland & Ylvisåker, 2015*; *Vanhelst et al., 2012b*; *Colbert et al., 2011*; *Vanhelst et al., 2012a*). Previous research has confirmed the acceptability of using these different models simultaneously in studies, as the levels of physical activity are comparable (*Vanhelst et al., 2012b*; *John, Tyo & Bassett, 2010*; *Lee, Macfarlane & Cerin, 2013*). Accelerometers were attached using an adjustable elastic waist belt above the right hip for seven consecutive days. Participants were asked to only remove the accelerometer while sleeping, and for water-based activities, such as swimming or bathing. Accelerometer counts were collected using 60-second epochs. Non-wear time, which was defined as $\geq 60$ min of consecutive zeros, was removed (*Choi et al., 2012*), and participants with less than five valid days of data (*i.e.,* at least 10 wearing hours)

were excluded from the analysis (*Grimm et al., 2012*). Although there is still no consensus on the optimal cut-points to estimate physical activity, the recommended cut points of Freedson (*Freedson, Melanson & Sirard, 1998*), and Evenson (*Evenson et al., 2008*), were used to convert the counts per minute into minutes of LPA (101 through 1,951 counts/minute were considered LPA in adults and older adults, and 101 through 2,296 counts/minutes were considered LPA in adolescents). By doing so, comparability with existing research is possible (*Gorman et al., 2014*; *Trost et al., 2011*; *Loprinzi et al., 2012*). The complete accelerometer data processing was performed using Actilife software version 6.

### Predictor variables: GIS-based neighborhood built environmental factors

Geographical data were obtained through the Service for Environmental Planning in Ghent. ArcGIS (version 10.3; ESRI) software was used to geocode participants' residences, to create the sausage buffers of 500 m and 1,000 m (1 km) around the home address for each participant, and to generate the included built environmental factors based on the International Physical Environmental Network (IPEN) guidelines (*Adams et al., 2014*). These guidelines provide standardized templates, including operational definitions, in- and exclusion criteria, and concrete examples of GIS-based computations. The templates are available in *Adams et al. (2014)* Sausage buffers are preferred over the more traditional Euclidian buffers, as sausage buffers are directly based on the road networks used to travel (*Forsyth et al., 2012*). The 1,000 m sausage buffer was chosen because this is considered to be a walkable distance (*i.e.,* within a 20 min walking distance) (*Adams et al., 2014*; *Cerin et al., 2016*), and because this scale is the most prevalent one in the physical activity literature (*Saelens et al., 2012*; *Oliver, Schuurman & Hall, 2007*; *Troped et al., 2010*). The 500 m sausage buffer was added as for older adults the near and direct environment might be more important due to mobility limitations (*Cerin et al., 2017*; *Carlson et al., 2012*). Five GIS-based neighborhood built environmental factors were included in the current study (see Table 1).

### Potential confounding/moderating variables: socio-demographic factors, valid days, and wear time

Age group, gender, educational level (primary, secondary, or tertiary), neighborhood SES, number of valid days, and wear time were selected a priori as potential confounding/moderating variables. Socio-demographic confounding variables were self-reported by the participants (or their parents) during the first home/school visit. Since adolescents were still studying, highest achieved educational level of the parent who filled in the questionnaire was included in the analyses as a proxy for their SES. Neighborhood SES was based on Belgian census income data from the National Institute of Statistics. Number of valid days and wear time were extracted from the accelerometer data.

## Statistical analyses

Descriptive statistics of participants' characteristics were calculated for the total sample and the three age groups (adolescents, adults and older adults) separately. Means and standard deviations were provided for normally distributed continuous variables, medians

**Table 1  Operational definitions for GIS-based built environmental factors.**

| Built environmental factor | Operational definition |
| --- | --- |
| Residential density | Number of residential dwellings (houses and apartments) fully or partially in the buffer divided by the residential land area (derived from residential parcels only) within participants' buffers. |
| Intersection density | Number of three- or more-way pedestrian-accessible street intersections divided by the area within participants' buffers. Intersections on limited access roads (*e.g.,* limited-access highways and on–ramps) were excluded. |
| Park density | Number of public parks of any size contained in or intersected by the buffer, divided by the land area within participants' buffers. A public park was defined as a government designated park of any size that was free of cost and open to the public and maintained by a government agency. Parks included improved or landscaped areas and unimproved or natural areas. |
| Public transport density | Number of bus, rail, or ferry stops and stations divided by the land area within participants' buffers. The complexity was shown by a variety of modes (ie, bus, rail, and ferry) and mode types (*e.g.,* regular bus vs bus rapid transit and light vs heavy rail) within and across cities. |
| Entropy index | The land-use mix diversity within the buffers, calculated using the following formula $H(S) = -\sum_{i=1}^{k}[(\rho i) \cdot (\ln \rho i)]/\ln k$ where $H(S)$, Entropy index (Shannon index); $\rho i$, the area of a particular category of land-use over the total area of all categories (within the scope of one district); and $k$, the number of land-use categories in the particular district (*Simons et al., 2017*). Six land use categories were included in the formula: residential, commercial, institutional, entertainment, food, and private/public recreational. |

and interquartile ranges for skewed continuous variables, and percentages for discrete variables. Linear mixed models were performed using the lmer() function of the lme4 package in R (v 4.1.0) to account for the nested structure of the data (*i.e.,* individuals were nested within neighborhoods) while examining the associations between GIS-based neighborhood built environmental factors and accelerometer-derived LPA (*Bates et al., 2014*). Firstly, a random intercept null model was fitted to estimate the variance in LPA explained at the neighborhood level. The intraclass cluster coefficient (ICC) was calculated from this model to estimate the proportion of total variance in LPA that could be attributed to neighborhood factors. Secondly, single-predictor models were run with each potential confounding variable (*i.e.,* age group, gender, educational level, neighborhood SES, number of valid days, and wear time), and each GIS-based neighborhood built environmental factor (*i.e.,* residential density, intersection density, park density, public transport density, and entropy) separately. Thirdly, multiple-predictor models were fitted including the significant confounding variables from the previous step, and the GIS-based built environmental neighborhood factors. Finally, the multiple-predictor models were extended with an interaction term (*i.e.,* age group*GIS-based neighborhood built environmental factor) to investigate the potential moderating role of age group. A likelihood ratio test was used to test the significance of the interaction terms by comparing models with and without interaction terms. All single- and multiple-predictor models were run separately for the environmental variables measured in a 500 m and 1 km sausage buffer. All analyses were performed in R (v 4.1.0) and the alpha level was set 0.05.

**Table 2  Descriptive statistics of the participants.**

| | Total sample (n = 1,652) | Adolescents (n = 150) | Adults (n = 1,059) | Older adults (n = 443) |
|---|---|---|---|---|
| **Socio-demographic variables** | | | | |
| Age, in years, mean (SD) | 48.8 (20.4) | 13.6 (1.3) | 42.9 (12.4) | 74.1 (6.2) |
| Gender, % female | 52.4 | 53.7 | 51.6 | 54.1 |
| Educational level | | | | |
| % primary | 9.9 | 4.8 | 4.2 | 25.2 |
| % secondary | 32.8 | 20.0 | 33.1 | 36.4 |
| % tertiary | 57.3 | 75.2 | 62.7 | 38.4 |
| **Accelerometer-derived variables** | | | | |
| LPA, min/day, mean (SD) | 304.6 (96.3) | 226.9 (46.8) | 335.2 (92.2) | 257.9 (84.3) |
| LPA, % of the day, mean (SD)[a] | 35.0 (10.5) | 27.6 (9.2) | 38.1 (10.3) | 30.1 (5.5) |
| Wear time | 869.4 (83.8) | 821.5 (54.0) | 882.3 (84.2) | 854.8 (83.0) |
| Valid days | 6.8 (0.8) | 7.4 (1.2) | 6.8 (0.8) | 6.8 (0.5) |
| **GIS-based neighborhood built environmental variables** | | | | |
| Residential density 500 m, mean (SD)[b] | 4770.5 (3226.2) | 4271.9 (3150.4) | 4795.6 (3227.7) | 4879.2 (3240.1) |
| Residential density 1,000 m, mean (SD)[c] | 4357.4 (2987.6) | 3875.4 (2669.5) | 4387.7 (3018.7) | 4448.3 (3006.0) |
| Intersection density 500 m, mean (SD)[d] | 160.5 (68.3) | 155.0 (63.0) | 159.9 (68.4) | 163.7 (69.6) |
| Intersection density 1,000 m, mean (SD)[e] | 157.2 (65.0) | 148.7 (56.6) | 158.1 (65.6) | 157.8 (66.1) |
| Park density 500 m, median (Q1–Q3)[f] | 8.0 (3.7–12.6) | 6.6 (0.0–13.3) | 7.6 (0.0–11.5) | 9.9 (5.5-15.8) |
| Park density 1,000 m, median (Q1–Q3)[g] | 7.2 (3.9–11.4) | 7.6 (3.7–12.5) | 6.9 (3.8–10.8) | 8.0 (4.5-13.3) |
| Public transport density 500 m, mean (SD)[h] | 33.7 (20.0) | 29.8 (19.8) | 34.4 (20.0) | 33.6 (20.0) |
| Public transport density 1,000 m, mean (SD)[i] | 32.9 (13.7) | 29.2 (14.4) | 33.1 (13.6) | 33.5 (13.4) |
| Entropy index 500 m, mean (SD)[j] | 0.5 (0.2) | 0.4 (0.2) | 0.5 (0.2) | 0.5 (0.2) |
| Entropy index 1,000 m, mean (SD)[j] | 0.5 (0.2) | 0.5 (0.1) | 0.5 (0.2) | 0.6 (0.2) |

**Notes.**

SD, Standard deviation; min, minutes; LPA, light-intensity physical activity.

[a] % of wear time.
[b] Number of dwellings per surface buffer 500 m.
[c] Number of dwellings per surface buffer 1 km.
[d] Number of intersections per surface buffer 500 m.
[e] Number of intersections per surface buffer 1 km.
[f] Number of public parks of all sizes per surface buffer 500 m.
[g] Number of public parks of all sizes per surface buffer 1 km.
[h] Number of public transportation stops of any kind per surface buffer 500 m.
[i] Number of public transportation stops of any kind per surface buffer 1 km.
[j] Range from 0 (=perfect homogenous land use) to 1 (=perfect heterogeneous land use).

# RESULTS

## Descriptive statistics of the participants

Participants with invalid accelerometer data (*i.e.,* less than five days with at least 10 h of wearing time) (n = 323) or GIS-data (n = 106) were excluded from the study. This resulted in a total sample of 1652 participants, of which 150 adolescents, 1059 adults and 443 older adults (see File S1). Descriptive statistics of the participants are presented in Table 2. The mean age of the sample was 48.8 (±20.4) years, and about half of the sample was female. The sample spent on average 304.6 (±96.3) min in LPA per day, ranging from 226.9 (±46.8) min/day for adolescents to 335.2 (±92.2) min/day for adults.

### Associations between GIS-based neighborhood built environmental factors and accelerometer-derived LPA

The ICC of the random intercept null model was 0.123, indicating that 12.3% of the variance in LPA can be attributed to the neighborhood level. Table 3 presents the results of the single- and multiple-predictor mixed effects regression models. Results of the multiple-predictor model for the 500 m sausage buffer showed that residential density is significantly related to LPA. Concretely, every increase of 1,000 dwellings per 500 m surface buffer was associated with a two minute decrease in LPA, given age group, gender, educational level and wear time are held constant. None of the other GIS-based neighborhood built environmental factors were related to LPA in the model with the 500 m sausage buffer. Results of the multiple-predictor model for the 1,000 m buffer showed that none of the included GIS-based built environmental factors (*i.e.,* residential density, intersection density, park density, public transport density and entropy) were significantly associated with LPA.

### Moderating role of age group on the associations between GIS-based neighborhood built environmental factors and accelerometer-derived LPA

Table 4 lists the results of the likelihood ratio test comparing the multiple-predictor model of accelerometer-derived LPA with and without interaction terms (GIS-based built environmental factors * age group). Results showed that none of the interaction effects were significant, indicating that the association between GIS-based built environmental factors and LPA did not differ depending on age group.

## DISCUSSION

This study is the first to investigate the association between GIS-based neighborhood built environmental factors and accelerometer-derived LPA in different age groups. The results showed only one significant association, namely that residential density (500 m buffer) was inversely associated with LPA in the total sample. This finding is in line the results of previous LPA studies, *Gonçalves et al. (2017)* and *Van Holle et al. (2014a)*, suggesting that residents of dense neighborhoods are less likely to engage in LPA. Important to note is, however, that the association was only present once the analyses were controlled for socio-demographic factors and wear time, and that the effect size of the observed association was rather limited. As such, the clinical relevance might be questioned.

At first sight, it seems that the current findings contrast with previous evidence regarding built environmental determinants of MVPA, showing that neighborhoods with high residential density were positively associated with MVPA (*Van Dyck et al., 2010*; *Cerin et al., 2018*; *Loh et al., 2019*). However, LPA and MVPA are interrelated as they both occur—together with sleep and sedentary behavior—within a finite 24-hour window (*Rosenberger et al., 2019*). Less time in one behavior might thus lead to more time in another behavior. Or, concretely, it might be that residents of more dense neighborhoods might partly replace their LPA by MVPA. More research including the full 24-hour activity cycle is

Compernolle et al. (2022), *PeerJ*, DOI 10.7717/peerj.13271

**Table 3  Associations between GIS-based neighborhood built environmental factors and accelerometer-derived LPA.**

| | Single-predictor models—LPA[j] | | | Multiple-predictor models—LPA[k] | | | | | |
| --- | --- | --- | --- | --- | --- | --- | --- | --- | --- |
| | | | | Buffer 500 m | | | Buffer 1,000 m | | |
| | B (SE) | *p* | 95% CI | B (SE) | *p* | 95% CI | B (SE) | *p* | 95% CI |
| **Socio-demographic variables** | | | | | | | | | |
| Adults (ref. adolescents) | **110.5 (8.1)** | **<0.001** | 93.43–124.69 | **88.8 (7.4)** | **<0.001** | 74.21–103.28 | **89.3 (7.5)** | **<0.001** | 74.64–103.99 |
| Older adults (ref. adolescents) | **32.4 (8.8)** | **<0.001** | 15.42–49.38 | 15.0 (8.1) | 0.06 | −0.93 to 30.85 | 16.4 (8.2) | 0.05 | 0.39–32.37 |
| Women (ref. men) | **18.48 (4.6)** | **<0.001** | 9.68–27.16 | **26.3 (4.1)** | **<0.001** | 18.34–34.23 | **26.4 (4.1)** | **<0.001** | 18.47–34.37 |
| Secondary (ref. primary) | **37.5 (8.4)** | **<0.001** | 20.95–54.06 | 4.7 (7.5) | 0.54 | −10.13 to 19.47 | 5.7 (7.5) | 0.45 | −9.06 to 20.53 |
| Tertiary (ref. primary) | **24.6 (8.3)** | **0.003** | 8.32–40.96 | **−18.1 (7.5)** | **0.02** | −32.76 to −3.47 | **−16.1 (7.5)** | **0.03** | −30.79 to −1.47 |
| **Neighborhood SES** | | | | | | | | | |
| High (ref. low) | 1.27 (11.59) | 0.913 | −22.04 to 24.24 | – | – | – | – | – | – |
| **Accelerometer-derived variables** | | | | | | | | | |
| Wear time (min/day) | **0.36 (0.02** | **<0.001** | 0.29–0.39 | **0.3 (0.02)** | **<0.001** | 0.26–0.36 | **0.3 (0.02)** | **<0.001** | 0.26–0.36 |
| Valid days | 4.6 (2.9) | 0.11 | −1.80 to 6.91 | – | – | – | – | – | – |
| **GIS-based neighborhood built environmental variables** | | | | | | | | | |
| Residential density 500 m[a] | −0.001 (0.001) | 0.30 | −0.003 to 0.001 | **−0.002 (0.001)** | **0.04** | −0.004 to −0.0005 | – | – | – |
| Residential density 1,000 m[b] | −0.001 (0.001) | 0.24 | −0.003 to 0.001 | – | – | – | −0.003 (0.002) | 0.09 | −0.01 to 0.0004 |
| Intersection density 500 m[c] | −0.05 (0.05) | 0.28 | −0.14 to 0.04 | −0.03 (0.05) | 0.58 | −0.12 to 0.07 | – | – | – |
| Intersection density 1,000 m[d] | −0.0007 (0.05) | 0.99 | −0.11 to 0.11 | – | – | – | 0.05 (0.07) | 0.51 | −0.09 to 0.18 |
| Park density 500 m[e] | −0.37 (0.29) | 0.21 | −0.94 to 0.21 | −0.04 (0.24) | 0.87 | −0.52 to 0.44 | – | – | – |
| Park density 1,000 m[f] | **−1.33 (0.61)** | **0.03** | −2.56 to −0.09 | – | – | – | −0.34 (0.52) | 0.51 | −1.36 to 0.72 |
| Public transport density 500 m[g] | −0.06 (0.14) | 0.67 | −0.33 to 0.21 | −0.11 (0.12) | 0.38 | −0.35 to 0.13 | – | – | – |
| Public transport density 1,000 m[h] | −0.15 (0.23) | 0.53 | −0.60 to 0.31 | – | – | – | 0.03 (0.02) | 0.90 | −0.42 to 0.48 |
| Entropy index 500 m[i] | −22.0 (13.6) | 0.11 | −49.95 to 1.88 | −13.5 (13.8) | 0.33 | −40.62 to 13.57 | – | – | – |
| Entropy index 1,000 m[i] | −22.2 (17.4) | 0.20 | −65.49 to 1.21 | – | – | – | −34.7 (19.8) | 0.08 | −73.51 to 4.19 |

**Notes.**

Min, minutes; LPA, light-intensity physical activity.

[a]Number of dwellings per surface buffer 500 m.

[b]Number of dwellings per surface buffer 1 km.

[c]Number of intersections per surface buffer 500 m.

[d]Number of intersections per surface buffer 1 km.

[e]Number of public parks of all sizes per surface buffer 500 m.

[f]Number of public parks of all sizes per surface buffer 1 km.

[g]Number of public transportation stops of any kind per surface buffer 500 m.

[h]Number of public transportation stops of any kind per surface buffer 1 km.

[i]Range from 0 (=perfect homogenous land use) to 1 (=perfect heterogeneous land use).

[j]Single-predictor models were run with each potential confounding variable (*i.e.,* age group, gender, educational level, neighborhood SES, number of valid days, and wear time), and each GIS-based neighborhood built environmental factor (*i.e.,* residential density, intersection density, park density, public transport density, and entropy) separately.

[k]Multiple-predictor models were fitted including the significant confounding variables from the previous step (*i.e.,* age group, gender, educational level, and wear time), and the GIS-based built environmental neighborhood factors.

Significant *P*-values (*i.e.,* ≤0.05) were displayed in bold.

**Table 4  Moderating role of age group on the association between GIS-based neighborhood built environmental factors and accelerometer-derived LPA.**

| Buffer size | Built environmental factor | Chi² (df) for interaction effect age group * built environmental factor[j] | p |
|---|---|---|---|
| 500 m | Residential density[a] | 2.05 (2) | 0.36 |
|  | Intersection density[b] | 0.04 (2) | 0.98 |
|  | Park density[c] | 1.89 (2) | 0.39 |
|  | Public transport density[d] | 2.29 (2) | 0.32 |
|  | Entropy index[e] | 0.89 (2) | 0.64 |
| 1 km | Residential density[f] | 1.70 (2) | 0.43 |
|  | Intersection density[g] | 0.05 (2) | 0.97 |
|  | Park density[h] | 0.43 (2) | 0.81 |
|  | Public transport density[i] | 0.50 (2) | 0.78 |
|  | Entropy index[e] | 0.92 (2) | 0.63 |

Notes.

[a] Number of dwellings per surface buffer 500 m.

[b] Number of intersections per surface buffer 500 m.

[c] Number of public parks of all sizes per surface buffer 500 m.

[d] Number of public transportation stops of any kind per surface buffer 500 m.

[e] Land-use mix diversity ranging from 0 (=perfect homogenous land use) to 1 (=perfect heterogeneous land use).

[f] Number of dwellings per surface buffer 1 km.

[g] Number of intersections per surface buffer 1 km.

[h] Number of public parks of all sizes per surface 1 km.

[i] Number of public transportation stops of any kind per surface buffer 1 km.

[j] Multiple-predictor models including age, gender, educational level, wear time, residential density, intersection density, park density, entropy index and the interaction term (*i.e.,* built environmental factor*age group).

Significant *P*-values (*i.e.,* ≤0.05) were displayed in bold.

needed in order to confirm the previous hypothesis, and to decide upon the most physical activity-friendly neighborhood environment.

Furthermore, it is important to shed light on the behaviors that are included in accelerometer-derived LPA to fully understand the inverse association. Although the interpretation of LPA is still not entirely clear, it seems that activities, such as gardening, cooking and cleaning belong to LPA (*Mendes et al., 2018*). As residents of neighborhoods with low residential density are more likely to live in large houses, with large gardens, this could possibly explain the inverse association (*Van Holle et al., 2014a*). Next to the behaviors that are classified as LPA, additional insight into the percentage of LPA that is spent walking in one's neighborhood could help to identify potential determinants. Given the limited number of observed associations between neighborhood built environmental factors and LPA, it is suspected that walking slowly in the neighborhood makes only a small contribution to the total time spent in LPA. This would also explain the disagreement between the current results and the ones previously found in built environment-walking studies. Several reviews found evidence of positive associations between residential density/urbanization, walkability, street connectivity, overall access to destinations/services, land use mix, pedestrian-friendly features and access to several types of destinations with walking (*Cerin et al., 2017*; *Hilland et al., 2020*; *Owen et al., 2004*). Studies combining GPS and accelerometer data are recommended to map the proportion of time in LPA spent in- and outdoors.
Contrary to our hypothesis, no stronger associations were observed between environmental neighborhood factors and LPA in adolescents and older adults, compared to young and middle-aged adults. Again, this could be explained by the fact that walking slowly in one's neighborhood represents only a small part of total accelerometer-derived LPA. If future research confirms that LPA occurs only to a limited extent in one's neighborhood, examining the role of the home, school and work environment will, just as with sedentary behavior, be more relevant to detect determinants than investigating the influence of neighborhood factors (*Compernolle et al., 2017*). Next to the home, school and work environment, psychological and social environmental factors were also understudied, and deserve more attention in future LPA studies.

This study has several strengths. To our knowledge, no previous studies have investigated the associations between GIS-based neighborhood built environmental factors and accelerometer-derived LPA in three different age groups. However, including different age groups is important in environmental research, as recommendations to policy makers and urban planners can only be formulated if an environment is supportive for all age groups. Secondly, GIS-based measures were used to assess neighborhood built environmental factors, and accelerometers were applied to estimate LPA. By relying solely on objective measures, recall and/or social desirability biases were eliminated, and accuracy of estimations was improved. Lastly, a sample of more than 1600 people was included in the current study, which guarantees sufficient power to detect potential associations. Limitations of the current study include firstly the lack of context- and location specific LPA information. Information on the context and location in which LPA was performed would have been helpful to interpret the results, and to formulate recommendations. Secondly, the cross-sectional design did not allow us to address the direction of causality. A longitudinal design would be recommended to understand the residential density-LPA association. Thirdly, the absence of information on sleep duration has prevented us from analyzing the so-called 24-hour activity behaviors (*i.e.,* sleep, sedentary behavior, LPA and MVPA) together. This would have been of added value as time spent in one behavior impacts the time spent in at least one of the other behaviors. Future studies should consider investigating all activity behaviors in the same model, using suitable statistical approaches, to formulate the most appropriate policy recommendations. Fourthly, the overrepresentation of highly educated persons, and adults in the sample limits the representativeness and generalizability. Moreover, the limited sample size for adolescents might have influenced the results, and have caused power issues. More than half of the adolescents were excluded from the analysis due to incomplete data. Specifically, 97 adolescents lack GIS data as they were recruited through schools and did not live in Ghent, and 126 adolescents did not provide valid accelerometer data. A more heterogeneous sample with regard to highest obtained educational level, and age group, would have improved the external validity of the results. Finally, the lack of universal consensus on accelerometer cut-points, especially in older adults, might have influenced the results of the current study. Time in LPA may have been overestimated for older adults with low fitness (*Del Pozo Cruz et al., 2021*).

## CONCLUSION

The current results suggest that the contribution of the objectively-measured physical neighborhood environment is limited in explaining accelerometer-derived LPA in all age groups. Only residential density (500 m buffer) was related to LPA. The small, but significant inverse association suggested that residents from dense neighborhoods are less likely to engage in LPA compared to residents from less dense neighborhoods. This association is opposite to the association between residential density and MVPA, and should be taken into consideration by policy makers and urban planners when designing health-enhancing neighborhoods. As no other associations were found between neighborhood built environmental factors and accelerometer-derived LPA, the role of the neighborhood environment seems negligible for LPA, and policy recommendations that were formulated based on MVPA research should be retained. More research is needed into the neighborhood influence on the full 24-hour activity cycle, and into the concept of accelerometer-derived LPA to fully understand the findings of the current study.

## ACKNOWLEDGEMENTS

The authors would like to thank the master students who contributed to the data collection, and the participants of the three studies (BEPAS Adults, BEPAS Seniors and IPEN adolescents).

### Funding

This research was supported by the Research Foundation Flanders (FWO) B/13018/01. Sofie Compernolle (FWO20/PDJ/088), Lieze Mertens (FWO17/PDO/140) and Jelle Van Cauwenberg (FWO 12I1117N) were supported by a postdoctoral fellowship of the Research Foundation Flanders. The funders had no role in study design, data collection and analysis, decision to publish, or preparation of the manuscript.

### Grant Disclosures

The following grant information was disclosed by the authors:
Research Foundation Flanders (FWO): B/13018/01.

### Competing Interests

The authors declare there are no competing interests.

### Author Contributions

- Sofie Compernolle analyzed the data, prepared figures and/or tables, authored or reviewed drafts of the paper, and approved the final draft.
- Lieze Mertens and Iris Maes analyzed the data, authored or reviewed drafts of the paper, and approved the final draft.
- Jelle Van Cauwenberg conceived and designed the experiments, performed the experiments, analyzed the data, authored or reviewed drafts of the paper, and approved the final draft.
- Delfien Van Dyck conceived and designed the experiments, performed the experiments, authored or reviewed drafts of the paper, and approved the final draft.

## Ethics

The following information was supplied relating to ethical approvals (*i.e.,* approving body and any reference numbers):

All three studies were approved by the Ethics Committee of the Ghent University Hospital (B670201423000) and all participants provided written informed consent.

## Data Availability

The raw data and the R script are available in the Supplementary Files.

## Supplemental Information

Supplemental information for this article can be found online at http://dx.doi.org/10.7717/peerj.13271#supplemental-information.

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
