# Peer review of "The association between Geographic Information System-based neighborhood built environmental factors and accelerometer-derived light-intensity physical activity across the lifespan: a cross-sectional study"

_PeerJ, doi:10.7717/peerj.13271_

## Round 0.1 · original submission · Major Revisions

Thank you for submitting the manuscript to PeerJ. It has been reviewed by experts in the field and we request that you make major revisions before it is processed further.

Please make the required changes, especially those of the reviewer 1.

We look forward to hearing from you soon.

Best wishes,

Badicu Georgian, Ph.D
PeerJ Academic Editor

Reviewer 1 ·

Excellent Review

This review has been rated excellent by staff (in the top 15% of reviews)
EDITOR COMMENT
The elaborate review is very detailed and constructive. Also, the detailed attention paid to statistical analysis is particularly useful. Congratulations to the reviewer for the professionalism of this report.

Basic reporting

Authors reported on a very interesting study on neighborhood built environmental (BE) factors and accelerometer-derived light-intensity physical activity (LPA). The manuscript is generally well written, and the rationale, objectives and methodology are clearly presented. The article adds to the currently scarce literature regarding accelerometer LPA outcome-GIS BE factors. The study objectives are promising and important for the field of health promotion. Also, investigating moderation between different age groups is an important step forward from the existing literature. Although I applaud the reasoning, objectives and the general quality/clarity of the manuscript, I have some major methodological concerns and other issues that the authors need to address before I can accept the manuscript for publication.

Abstract
• Authors should consider changing the word “correlates” to “association” in the first line of the background section of the abstract
• Authors should be specific with what they mean by “A small but significant association was observed between residential density” in the results section of the introduction. Is it a positive or negative association? Please clarify
• I am a little bit confused with the statement “demonstrating that every decrease of 1000
dwellings per surface buffer was associated with a two-minute increase in LPA”. Is the Unit of LPA in minutes? Authors should please clarify.

• Is the interpretation of the beta coefficient correct in the first line of the result section in the abstract? “A small but significant association was observed between residential density (500 m
buffer) and LPA in the total sample (B=-0.002; SE=0.0001; p=0.04), demonstrating that every
decrease of 1000 dwellings per surface buffer was associated with a two minute increase in LPA.” If the beta coefficient is negative, the interpretation is that for every 1-unit increase in the predictor variable (i.e , residential density ),the outcome variable(i.e.LPA) will decrease by the beta coefficient value. Based on this, I was expecting the authors to write “……..demonstrating that every increase of 1000 dwellings per surface buffer was associated with a two minute decrease in LPA” Can the authors clarify this?

• Authors should mention the number of days the accelerator/GPS was worn in the methods section of the Abstract
Background
• [line 58] Please provide a citation
• Can the authors maybe provide some more background on the importance of the different moderating variables analyzed ( i.e.age group-adults, adolescents,-etc) in relation to BE-LPA association . A paragraph will be Okay
• [line 67] please provide a citation
• The rational for examining the five BE should mentioned in the background section. Why did the authors choose to examine entropy index? Why did the authors choose to examine residential density and not other BE factors such as water bodies? At least the reader should be given a background on how each of the five BE factors are important to LPA.

Experimental design

Participants and procedure
• Authors should consider changing the sub heading title “Participants and procedure” to something along the lines of “Study participant selection procedure”
• [line 112] how was “low” and high walkability defined? The authors need to explain the criteria for “low” and “high” walkability. The same applies to the SES variable. What was the criteria for selecting a neighborhood as low or high SES? The authors need to clarify
• Authors should consider including a flowchart to summarize the participant selection process (The number of participants that were dropped or excluded at each stage) . This flowchart can be put in the appendix

Outcome variable: LPA

• Were the accelerometers equipped with GPS? If yes authors did not provide any details regarding raw GPS data processing. Was there malfunctioning due to signal quality when exposed to different environments? How did you deal with that? How about signal loss indoor? Did it occur? If you are limited in pages, at least an appendix or consort diagram of N numbers needs to be provided. Without knowing these details, it is tough to evaluate the quality of GPS data before aligning with accelerometer by timestamps.
• When was data collected? Did they have lots of non-wearing hours?
• The authors fail to mention the number of bouts they used for their accelerometer processing. Did they use a 5-min bout, 10-min bout? Please clarify
• Transparency and quality of the spatial and statistical analyses can be improved by providing the reader with a flowchart, depicting compliance and loss of data in each step of the analytical process. The authors should provide information summarizing data loss at each step of the accelerometer data
• There are doubts for readers about the level at which data are in the end analyzed. Overall you need to clarify what is the unit of analysis. Was the data analyzed at 60 -second epoch collected by the accelerometer? Was the data cumulated to 1 minute? All these need to be spelled out clearly in the methods section
• It is not clear if the authors applied a point-based method to access cumulative exposure and examine its cross-sectional associations with LPA outcomes. Authors should clarify if they used a point by point base approached and mention the limitations that come with it in the discussion section.

Predictor variables: GIS-based neighborhood built environmental factors
• Why did you select these 5 GIS-based neighborhoods built environmental factors? In other words, why they made the BE set but not others? Were they examined most frequently by previous studies? Were they related to LPA outcomes for certain reasons? The rationale should be provided.
• Why were 500 m and 1000 m (1 km) around the home addresses of all participants selected? Are these arbitrary or are there specific reasons. Authors need to justify their reasoning for selecting these buffer radii.
• Authors fail to go into details on the methodologies used to derived BE-factors. Please elaborate and explain the data-extraction from GIS-registries. What were the data sources? What software was used to perform your GIS operations? All of this information are missing in the methods section.

• What was the logical reasoning for restricting your analysis to points within your 500 or 1000 buffer radius? Why ignore GPS points that fall out the home buffer. Light intensity physical activity may also occur outside the selected buffer radii. If the authors are restricting to only points that fall within the buffers, they are not using the full dataset over the 7 day period.
• Authors should consider as a form of sensitivity analysis, perform the same analysis but this time using only points that fall outside the buffer radius. They can then compare the results to the results of the points that fall inside the buffer radii.
• Authors should consider adding a table to illustrate the BE factor assessment method / operationability and data sources similar to table 1 of Troped et al 2010 . This will make it easier for non-spatial readers to grasp how your GIS BE factors were derived.
• It is not clear how the authors merged or linked the accelerometer data to the GIS built environmental factors. Authors need to explain into detail how they went about this process. At what unit or level was this done. Authors should please clarify how this linkage was performed. Was a GPS point linked to the closest built environment factor? Was there a spatial join? Please clarify
• At what level was the analysis performed? Did the authors apply a point-based method to access cumulative BE factor and use that to examine its cross-sectional associations with LPA outcome in the seven days.? Authors need to clarify this in their writeup. If this was their approach its limitation must be discussed in the discussion section.

Validity of the findings

• Authors are encouraged to revisit their rationale (and potentially also their methodology) about conceptualizing environmental factors and elaborate in the discussion-section on this concept.
• Authors should consider reporting the output as per IQR increase
• Transparency and quality of the spatial and statistical analyses can be improved by providing the reader with a flowchart, depicting compliance and loss of data in each step of step of the analytical process.
• I would like to see more policy implication of your results
Table 1
• Authors should consider not reporting the range of the age in the first row of the age variable. Just report the mean and SD

Table 2
• please add a foot note explaining what variables were included in the single predictor and multiple predictor models
• Authors should report the confidence intervals in addition to the p-values
Table 3
Authors should add footnotes to table 3
-explaining to the reader how the the what variables were included in the Chi2 (df) for interaction effect
- at what alpha level( eg, <0.05)? is the p-value significant. Please make sure to include this in the table

overall, tables need to be reformatted for easier reading and clarity. Please add foot notes to all your tables where appropriate

Reviewer 2 ·

Basic reporting

no comment

Experimental design

no comment

Validity of the findings

no comment

Additional comments

Thank you for the opportunity to review the paper on accelerometer-based light-intensity physical activity, with the strength of incorporating three cohort databases. This is an important paper looking at the GIS-based environmental correlates of light-intensity physical activity. The findings can offer some insights into the target population at risk of physical activity promotion. To improve the paper, I have several comments for the authors to consider, as follows -

1. The paper would be improved by a better justification for the need to investigate the moderating effect of age in the Introduction section. There is a comprehensive massive transport system in Belgium, so the mobility may not that limited to adolescents and older adults although they could not drive cars. Additionally, the authors can consider driver license or access to public transport as a confounding factor in the models if such information is available.

2. In the single-predictor models, residential density within 500 meters was not associated with light-intensity physical activity; however, it became significantly associated with light-intensity physical activity in the multiple-predictor models, after adjusting for other confounding factors. This significant association between residential density and light-intensity physical activity needs to be interpreted in caution.

·

Basic reporting

I would suggest the authors to explore more on the literature that support why additional study utilizing accelerometer accessed LPA is needed. Please check all recent published manuscripts.
While the authors do a good job of highlighting pre-existing work as it relates to built environmental factors and LPA, as well as the potential modification effect of age, it would be helpful to more clearly emphasize why the authors are interested in examining the relationship/moderation (beyond the fact that it has never been done before). Specifically, self-reported walking date focus on walk, while accelerometer assessed LPA might include all kinds of activity fall into the cut-off points. How comparable are the current results versus previous studies (line 249, gardening, cooking and etc.)? In addition, effect of age may not have been examined in one paper, but different manuscripts using the same data source may have explored the association between different age groups. (e.g., https://ijbnpa.biomedcentral.com/articles/10.1186/s12966-019-0886-2)

Experimental design

Another suggestion was to explore MVPA, LPA, SED and sleep (compositional data analysis through isometric log-ratio) at the same time. As the authors already mentioned in the discussion, it is necessary to explore different PA intensity while considering their association within 24 hours window. Looking at LPA alone didn’t tell the whole story, hence limited the depth of the current study.

Validity of the findings

The strength mentioned are not very convincing. Regarding the age group, it is good intention to study age differences on the association, the sample size difference should not be ignored. Also, it would be hard for policy maker to accommodate different needs across different age groups within same neighborhood, additional thoughts should be put into the discussion. Line 271-273, several studies have examined associations between built environmental factors and accelerometer measured PA (BI Chen et. al., 2019; D Salvo et. al., 2011; AK Solbraa et. al., 2018; PJ Troped et. al., 2010 to name a few). In addition, self-reported measure does have its limit, but accelerometer measure (on waist) also has its limitations on accuracy and tracking activities such as weight-lifting, casual bike riding and etc. where movements on the waist are hard to detect and track.

Additional comments

1. Line 104-105, any potential participants in both BEPAS and BEPAS Seniors study? Any impact on the result?
2. Line 136, since different accelerometers were used, any calibration/adjustment on the data to make results comparable?
3. Line 144, please provide more details on Freedson and Evenson cut-off points, and why authors choose to use each on specific age groups (add ref).
4. Line 154, any data on average age in the area? Might want to add as control/covariates when exploring age groups.

---

## Round 0.2 · Minor Revisions

Thank you for submitting the manuscript to PeerJ. Great improvements were performed in the manuscript. Currently, the article is acceptable for publication with minor revisions.

The only question for authors is why more than half of the adolescents were excluded (i.e., 150 included in the analysis out of 373 enrolled in the study)? I think that there are many people excluded in the conditions in which their total is 373 participants, please detail this aspect very clearly.

Reviewer 1 ·

Basic reporting

Authors have addressed all my concerns adequately

Experimental design

Authors have addressed all my concerns adequately

Validity of the findings

Authors have addressed all my concerns adequately

Additional comments

Authors have addressed all my concerns adequately

Reviewer 2 ·

Basic reporting

I thank the authors for responding to each of my comments. The revised paper has been greatly improved by clarifying the rationale of the investigation into the potential moderation effect and providing more details of the Methods and Discussion section.

Experimental design

The selection of participants was clearer after adding a flow chart. The only concern is that more than half of the adolescents were excluded (i.e., 150 included in the analysis out of 373 enrolled in the study), which has been added to the revised paper as a limitation. I have no further comments on the Method section.

Validity of the findings

The results of this study showed no strong association between environmental attributes and light-intensity physical activity, with only one plausible association of 500-m residential density with light-intensity physical activity after adjusting for sociodemographic characteristics. It is suitable to focus on the importance of taking 24-hour behaviours into account in future research, as the authors have done, rather than focus on the plausible 'negative' associations observed in the Discussion section. I have no further comments.

---

## Round 0.3 · accepted · Accept

I am writing to inform you that your manuscript has been Accepted for publication.

Congratulations!

Badicu Georgian, Ph.D
Academic Editor, PeerJ